# Spatial regulation of contractility by Neuralized and Bearded during furrow invagination in *Drosophila*

Gantas Perez-Mockus[1,2,3], Khalil Mazouni[1,2], Vanessa Roca[1,2], Giulia Corradi[1,2], Vito Conte [4] & François Schweisguth[1,2]

Embryo-scale morphogenesis arises from patterned mechanical forces. During *Drosophila* gastrulation, actomyosin contractility drives apical constriction in ventral cells, leading to furrow formation and mesoderm invagination. It remains unclear whether and how mechanical properties of the ectoderm influence this process. Here, we show that Neuralized (Neur), an E3 ubiquitin ligase active in the mesoderm, regulates collective apical constriction and furrow formation. Conversely, the Bearded (Brd) proteins antagonize maternal Neur and lower medial–apical contractility in the ectoderm: in *Brd*-mutant embryos, the ventral furrow invaginates properly but rapidly unfolds as medial MyoII levels increase in the ectoderm. Increasing contractility in the ectoderm via activated Rho similarly triggers furrow unfolding whereas decreasing contractility restores furrow invagination in *Brd*-mutant embryos. Thus, the inhibition of Neur by Brd in the ectoderm differentiates the mechanics of the ectoderm from that of the mesoderm and patterns the activity of MyoII along the dorsal–ventral axis.

[1] Department of Developmental and Stem Cell Biology, Institut Pasteur, F-75015 Paris, France. [2] CNRS, UMR3738, F-75015 Paris, France. [3] Univ. Pierre et Marie Curie, Cellule Pasteur UPMC, F-75015 Paris, France. [4] Institute for Bioengineering of Catalonia, Barcelona Institute of Science and Technology, 08028 Barcelona, Spain. Correspondence and requests for materials should be addressed to V.C. (email: vconte@ibecbarcelona.eu) or to F.S. (email: fschweis@pasteur.fr)

During development, morphogenesis arises from mechanical forces generated by actomyosin contractility[1,2]. Genetic information provides the spatial and temporal cues that pattern actomyosin contractility, hence generating shapes in a precise and reproducible manner[3]. Understanding how genes regulate force generation and possibly modulate material properties during morphogenesis are major challenges.

Gastrulation in the early *Drosophila* embryo is an outstanding model to study how contractility regulates epithelial morphogenesis at the embryo scale[2–7]. At the onset of gastrulation, the embryo consists in a single sheet of ~5000 cells. Dorsal–ventral (DV) patterning of the embryo culminates into the activation of a G protein-coupled receptor (GPCR)-Rho signaling cascade in ventral cells[8–13]. This leads to the recruitment and activation of

**Fig. 1** Ectopic Brd inhibits apical constriction. **a** Genetic control of Neur activity. Twist positively regulates *neur* gene expression in the mesoderm, whereas Snail represses the expression of the *Brd* family genes. Brd proteins antagonize Neur. **b** Brd activity (pink) is restricted to the ectoderm by Snail. Inhibition of Neur (orange) by Brd restricts the activity of Neur (green) to the mesoderm. **c–e** Live imaging of wild-type (**c**), *mat>mα+Tom* (**d**), and *sna>Brd^R* (**e**) embryos expressing MyoII-GFP (not shown) and Gap43-Cherry (white; tracking, red). Surface area values were measured for cells undergoing invagination (blue in **c–e**). In this and all other figures, time (*t*) is in minute (min) after the onset of apical constriction (used as *t* = 0). **f–h** Analysis of apical constriction. Surface value distributions highlight the rapid and synchronous constriction of ventral cells in wild-type embryos (**f**). Each plot corresponds to a single embryo (see Supplementary Fig. 1c and d for the analysis of multiple embryos). Each dot represents one cell (mean values shown as a blue curve; control wild-type curve is red in (**g** and **h**). Maternal over-expression of *mα* and *Tom* (**g**) as well as zygotic expression of *Brd^R* in the mesoderm (**h**) delayed furrow formation and led to the persistence of large non-constricting cells during furrow formation. In this and all other figures, scale bar is 10 μm

Myosin II (MyoII) in a DV gradient[14–16], resulting in the formation of contractile actomyosin meshworks[6,14]. At the cellular level, these cortical actomyosin meshworks undergo cycles of assembly and disassembly at the medial–apical cortex, thereby generating contractility pulses[6,18]. At the tissue-scale, these meshworks are interconnected via E-Cadherin to form a supracellular network, thereby ensuring tight mechanical coupling of the ventral cells[14,19]. Contraction of this supracellular tensile network causes rapid and collective apical constriction of the ventral cells[6,14]. At the embryo scale, the geometry of the force-generating domain orients the pattern of force within each cell and defines the shape and position of the ventral fold[7,20,21]. In addition, temporal correlations between the rates of apical constriction and cell elongation suggest that apical constriction

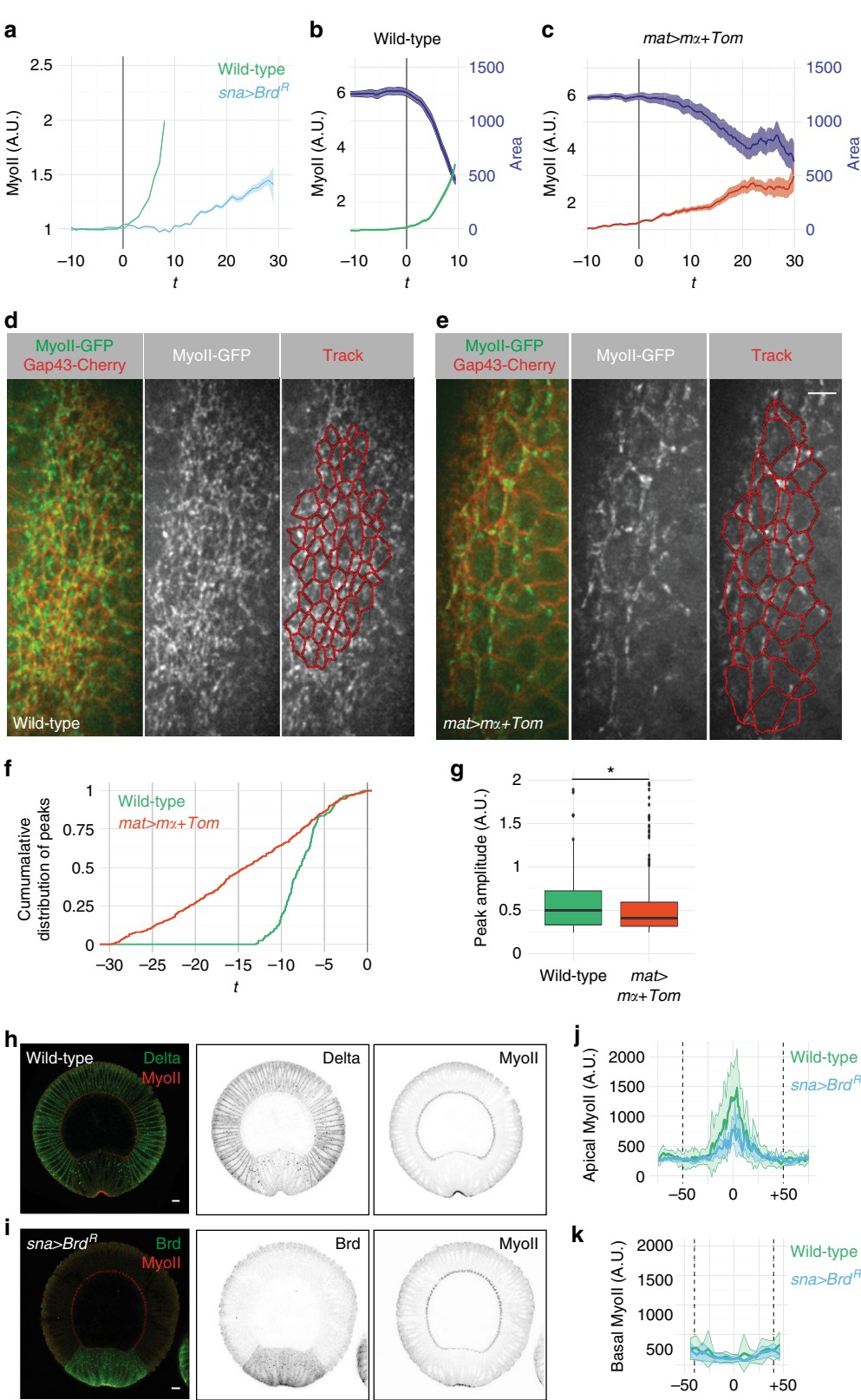

causes cell lengthening[22,23]. Once elongated, ventral cells shorten along their apical–basal axis, possibly as the result of decreasing stiffness at their basal cortex, and this shortening is thought to drive furrow invagination[24]. As the ventral furrow invaginates, ectodermal cells flow towards the ventral midline[7]. Although these studies have provided remarkable insights into force generation and shape changes within the mesoderm, how ectoderm cells respond to these forces and whether their contractility is regulated during gastrulation remain largely unknown[7].

The *Drosophila Bearded* (*Brd*) genes encode a family of small proteins (collectively referred here as *Brd*) that interact with the E3 ubiquitin ligase Neuralized (Neur) and inhibit its activity by competing substrates for binding Neur[25–30]. During gastrulation, the *Brd* genes are expressed only in the ectoderm, due to their repression by Snail in the mesoderm[25], whereas Neur is expressed both maternally and zygotically (Fig. 1a). As Brd inhibits Neur, the latter is only active in the mesoderm where it regulates the endocytosis of the Notch ligand Delta[25,26,31,32] (Fig. 1b). Whether Neur and Brd have additional functions during gastrulation is unknown.

Here, we find that Neur is required in the mesoderm to promote the recruitment of MyoII at the apical cortex of ventral cells, collective apical constriction, and rapid furrow formation, whereas Brd is critically required in the ectoderm to lower contractility and permit furrow invagination. We propose that the inhibition of Neur by Brd in the ectoderm establishes a mechanical difference between mesoderm and ectoderm and that this embryo-scale regulation of contractility is necessary for furrow invagination.

## Results

**Ectopic Brd delays furrow formation**. We previously showed that the early and ubiquitous expression of the Brd family members Enhancer of split mα (mα) and Tom was sufficient to perturb ventral furrow formation. This raised the possibility that Neur, the known target of mα and Tom, regulates this process[33]. However, as this observation was based on fixed embryos, it remained unclear how these Brd proteins interfered with this dynamic process.

Here, we first used live imaging to monitor the dynamics of apical constriction in embryos expressing a membrane-targeted RFP (Gap43-Cherry) and GFP-tagged MyoII (MyoII-GFP). In wild-type embryos, furrow invagination was observed ~6–7 min (min; $n = 8$) after the onset of constriction ($t = 0$). Analysis of apical constriction in the ~6 ventral-most cells confirmed that these cells undergo rapid and collective constriction (Fig. 1c, f; Supplementary Movie 1 and Supplementary Fig. 1a–d)[4–6]. As noted earlier[14], ventral cells constrict apically in an anisotropic manner, as shown by an increase in their aspect ratio

(Supplementary Fig. 1a, b). By contrast, over-expression of mα and Tom using a maternal Gal4 driver (*mat>mα+Tom*) delayed furrow formation and invagination (Fig. 1d, g; Supplementary Movie 1). Also, ventral cells underwent apical constriction asynchronously, with some cells constricting while others failed to constrict, and over a longer time period (Supplementary Fig. 1c, d). This defect in collective apical constriction, with cells constricting individually and asynchronously, is similar to those seen in *folded gastrulation*[5] and *concertina* mutants[34], or upon light-induced inhibition of contractility[35], suggesting that this distribution of constricting cells results from reduced contractility[36].

To test whether expression of Brd only in the mesoderm was sufficient to cause these defects, a stabilized version of Brd, Brd[R], was expressed conditionally in the mesoderm using the *snail* (*sna*) promoter. Conditional expression was achieved by introducing a GFP reporter and a transcriptional stop signal, flanked by two FRT sites, upstream of the *Brd[R]* gene so that it was expressed in the mesoderm only upon FLP-mediated excision of the stop cassette (Supplementary Fig. 2a–e). Live imaging of *sna>Brd[R]* embryos showed that ectopic Brd in the mesoderm interfered with collective apical constriction and delayed furrow invagination (Fig. 1e, h; Supplementary Movie 1 and Supplementary Fig. 1c, d). We conclude that Brd can delay furrow formation by antagonizing its target(s) in the mesoderm.

**Ectopic Brd inhibits apical contractility**. We next examined the effect of ectopic Brd on MyoII levels using live imaging of embryos expressing Gap43-Cherry and MyoII-GFP. As reported earlier[6,18], MyoII levels progressively increased at the medial–apical cortex as ventral cells constrict their apices in wild-type embryos (Fig. 2a, b). By contrast, both maternal and mesoderm-specific expression of Brd significantly slowed down the recruitment of MyoII in the mesoderm (Fig. 2a, c). In addition, the myosin network appeared less dense (Fig. 2d, e; Supplementary Movie 2). To further characterize the effect of Brd on MyoII dynamics, we studied the cycles of MyoII assembly–disassembly at the medial–apical cortex. In wild-type embryos, ventral cells underwent 2–3 pulses per cell within a ~10 min period preceding invagination ($2.4 \pm 1.2$, $n = 79$ cells; Supplementary Fig. 3c, e). By contrast, pulses varied greatly from cell to cell in *mat>mα+Tom* embryos (Supplementary Fig. 3d, f–h) and while more pulses were scored per cell ($4.5 \pm 2.2$, $n = 86$ cells), these were distributed over a ~30 min period (Fig. 2f). In addition, ectopic Brd reduced the amplitude of these pulses (Fig. 2g). Thus, ectopic Brd appeared to interfere with the recruitment of MyoII at the medial–apical cortex of ventral cells and this correlated with weaker pulses distributed over a longer period of time prior to invagination.

**Fig. 2** Ectopic Brd affects recruitment of apical MyoII. **a** Time-course accumulation of apical MyoII (total intensity in arbitray units (A.U.)) in ventral cells of one representative wild-type (green, $n = 3$) and *sna > Brd[R]* embryos (blue, $n = 3$). Mean and s.e.m. values are shown (see complete data set in Supplementary Fig. 3a). In this and other panels, the onset of apical constriction ($t = 0$) is indicated with a gray bar. **b, c** Apical MyoII-GFP levels (green) increased sharply as ventral cells constricted (surface area in pixels, blue) in wild-type embryos (**b**). Over-expression of *mα* and *Tom* led to slow MyoII increase and delayed constriction (**c**). One representative embryo is shown (see complete data set in Supplementary Fig. 3b). **d, e** Snapshots of MyoII-GFP (green) and Gap43-Cherry (red) movies of wild-type (**d**) and *mat>mα+Tom* (**e**) embryos. A dense myosin meshwork running across cell–cell boundaries was observed in wild-type embryos (**d**), whereas apical MyoII organized in a loose network in *mat>mα+Tom* embryos **e**. Anterior is up. **f, g** Cumulative distribution of MyoII pulses (**f**) and box plots of pulse amplitude values (**g**) in wild-type ($n = 190$ pulses, 79 cells in three embryos over 10.3 min) and *mat>mα+Tom* embryos ($n = 390$, 85 cells in three embryos over 33.6 min). MyoII pulses just precede furrow invagination in wild-type embryos, whereas these were distributed over ~30 min in *mat>mα+Tom* embryos (**f**). Also, higher amplitude pulses were seen in wild-type embryos (**g**, $p = 0.004$). In these and all other Tukey box plots, whiskers extend to the highest and lowest values that are within 1.5 inter-quartile range and dots are outlier values. **h, i** Cross-section views of wild-type (**h**) and *sna>Brd[R]* (**i**) early stage six embryos. Endocytic Dl (**h**) and Brd[R] (HA tag, **i**) marked the mesoderm. MyoII (red) accumulated at the apical cortex of ventral cells. **j, k** Quantification of apical and basal MyoII in wild-type (green, $n = 7$) and *sna>Brd[R]* embryos (blue, $n = 3$) at early stage 6. Ectopic Brd led to low apical MyoII levels but had no detectable effect on basal MyoII (x axis: distance to the midline, $x = 0$; limits of mesoderm defined by Dl or Brd[R] are indicated with dotted lines)

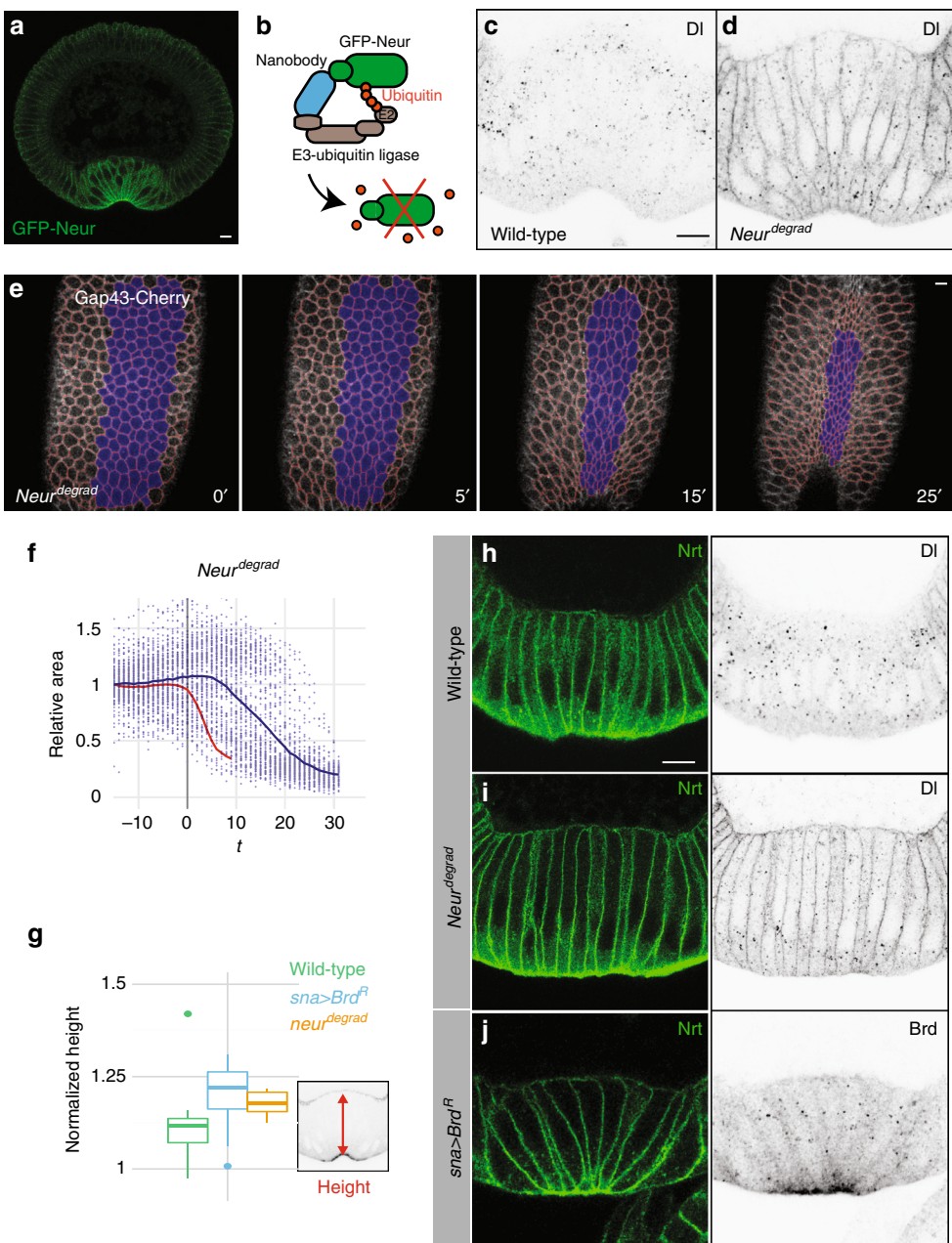

**Fig. 3** Depletion of Neur results in defective apical constriction. **a** Cross-section view of a GFP-Neur embryo showing apical accumulation of Neur (GFP, green; four copies of the BAC transgene; $n = 2$ experiments) in ventral cells during furrow formation. **b** deGradFP strategy: GFP-Neur (green) was targeted for degradation using a Cullin-based E3 ubiquitin ligase complex via an anti-GFP nanobody (blue). **c, d** Cross-section views of wild-type (**c**) and $neur^{degrad}$ (**d**) embryos. In wild-type embryos (**c**), Dl accumulates into endocytic vesicles specifically in the mesoderm. In 60% of the $neur^{degrad}$ embryos ($n = 35$), Dl was mostly detected at the membrane, indicative of a strong Neur depletion (**d**). **e** Live imaging of a $neur^{degrad}$ embryo expressing Gap43-Cherry (white; ventral cells in blue, tracking in red). A delay in furrow formation was observed. **f** Apical constriction (quantified as in Fig. 1f) was delayed in $neur^{degrad}$ embryos relative to wild-type embryos (red). Intermingled constricting and non-constricting cells were observed ventrally (Supplementary Fig. 4l). **g-j** Ventral cell lengthening was measured prior to furrow formation (stage I, as defined in Sweeton et al.[5]; $n = 3$ experiments) in wild-type (**g, h**; $n = 9$), in $neur^{degrad}$ (**g, i**; $n = 5$) and $sna>Brd^R$ embryos (**g, j**; $n = 14$) stained for Nrt (**h-j**) and Dl (**h, i**), or ectopic $Brd^R$ (**j**; HA epitope). Ectopic Brd accumulated at the apical cortex and in cytoplasmic dots (**j**). Lengthening increased upon inhibition and depletion of Neur (**g**)

To further test the role of ectopic Brd in MyoII recruitment, we next quantified the level of MyoII in cross-section views of wild-type and $sna>Brd^R$ embryos at mid stage 6 (Fig. 2h–k). We found that apical MyoII levels were reduced in $sna>Brd^R$ embryos with a ~3-fold decrease in peak levels (Fig. 2j). By contrast, basal MyoII levels were not detectably changed (Fig. 2k). This suggests that defective invagination may result from decreased apical contractility rather than from altered basal stiffness. Finally, like

$mat>ma+Tom$ embryos[33], we observed that $sna>Brd^R$ embryos exhibited a flat ventral surface.

Altogether, our results indicate that ectopic Brd inhibits apical contractility, leading to defective collective apical constriction and delayed furrow formation. We therefore propose that ectopic Brd inhibits a positive regulator of apical contractility in the mesoderm.

**Neur is required for apical constriction**. The E3 ubiquitin ligase Neur is the only known target of the Brd proteins[25,26]. It is expressed both maternally and zygotically under the control of Twist, predicting higher Neur levels in the mesoderm. Using functional GFP-tagged Neur (GFP-Neur; Supplementary Fig. 4a–c), we were able to detect Neur in ventral cells where it accumulated apically (Fig. 3a). If ectopic Brd acts by inhibiting Neur in the mesoderm, then a loss of *neur* activity should mimic the *Brd* over-expression phenotype. However, the zygotic loss of *neur* showed no ventral furrow phenotypes, possibly due to compensation by maternal *neur*[33]. As it was not possible to

obtain eggs from *neur* germline clones due to the essential function of *neur* in the female germline[37], we used the deGradFP method[38] to deplete the maternal and zygotic pools of Neur. This method relies upon a genetically encoded anti-GFP nanobody (Vhh) fused to the recognition subunit of a Cullin-based E3 ubiquitin ligase complex (Slmb). Conditional expression of this Slmb–Vhh fusion protein results in the degradation of GFP-tagged proteins (Fig. 3b). We applied this method to Neur by expressing Slmb–Vhh in the germline of *trans*-heterozygous *neur* mutant females rescued by GFP-Neur. These *neur*[degrad] females lacked endogenous Neur and only expressed GFP-tagged Neur.

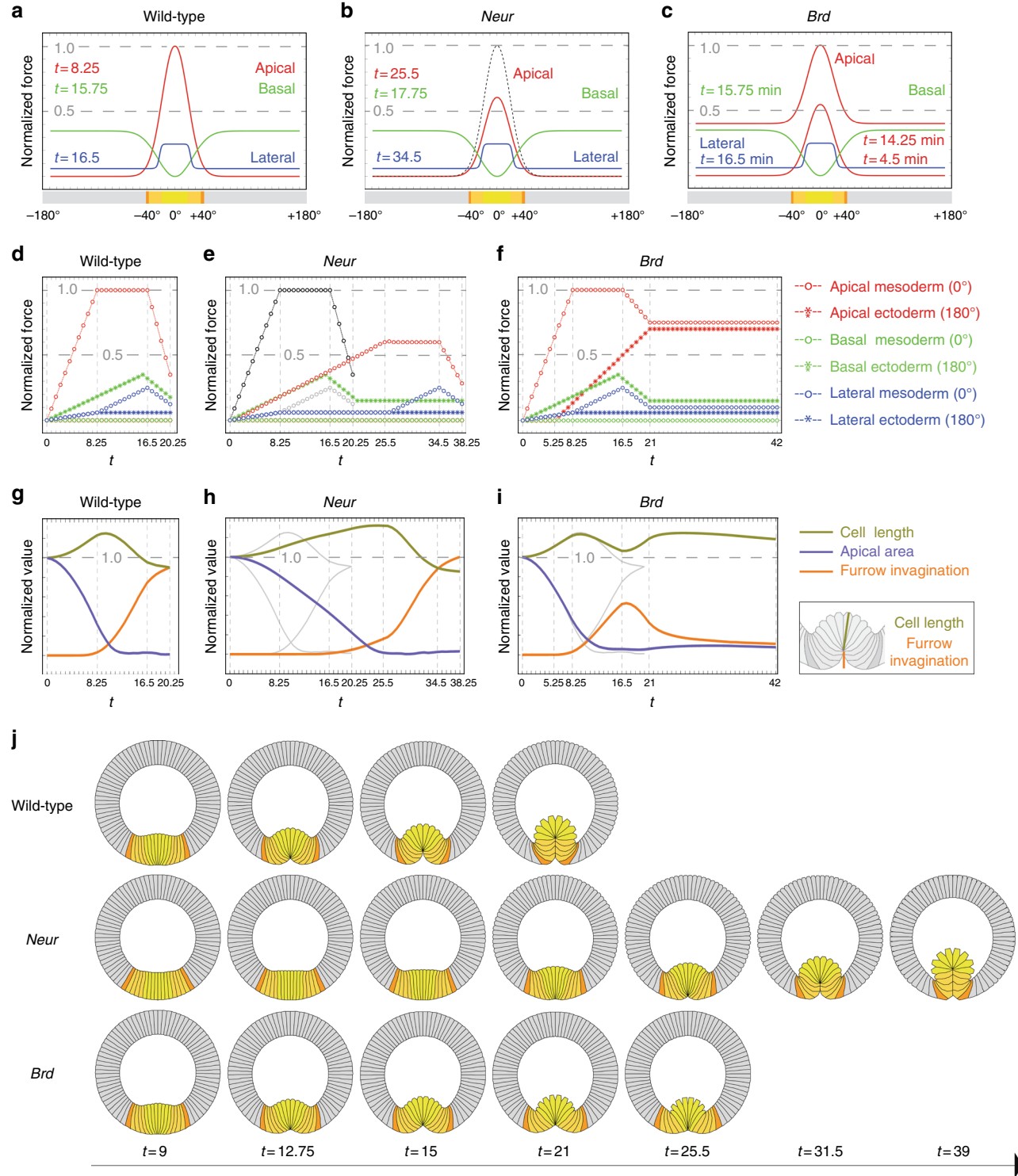

They laid eggs for about a week before becoming sterile. Interestingly, sterility correlated with oogenesis defects associated with defective Dl-Notch signaling (Supplementary Fig. 4d–f). The embryos laid by young *neur^degrad* females showed defects in Dl endocytosis in the mesoderm at early stage 6 (Fig. 3c, d) indicative of Neur depletion. Thus, Neur levels appeared to be optimally reduced down to a threshold level at which no egg is produced.

We then used live imaging to further study furrow formation in *neur^degrad* embryos expressing the Gap43-cherry marker (Fig. 3e; Supplementary Movie 3; see *control^degrad* embryos in Supplementary Fig. 4g). We first found that the duration time of furrow invagination was variable amongst the progeny of the *neur^degrad* females with about a third of these embryos significantly deviating from the wild-type controls (Fig. 3f; Supplementary Fig. 1c). This variability possibly resulted from differences in zygotic *neur* and/or residual levels of GFP-Neur that appeared to decrease over time in the germline. In the *neur^degrad* embryos that showed delayed invagination, we observed that only a subset of the ventral cells constricted and that constricting cells were intermingled with non-constricting cells over a broad ventral domain (Fig. 3e, f; Supplementary Fig. 1d). This phenotype was very similar to the one seen in *mat>mα+Tom* and *sna>Brd^R* embryos (Fig. 1d, e, g, h) and analysis of apical constriction along the DV axis revealed a decrease in the constriction gradient in these three genotypes (Supplementary Fig. 4h–k, n). We therefore conclude that Neur is required for collective apical constriction. Also, the phenotypic similarities between embryos with depleted Neur and those with ectopic Brd strongly indicated that Neur is the target of Brd in the mesoderm.

We next studied furrow formation in fixed embryos at stage 5–6. Upon depletion of Neur, a fraction (10/46) of the *neur^degrad* embryos deviated from wild-type embryos in that they displayed a flat ventral surface (Fig. 3h, i) like in *sna>Brd^R* embryos (Fig. 3j) with apical MyoII (Supplementary Fig. 4l, m). Despite this defect in furrow formation, ventral cell lengthening was observed in both *neur^degrad* and *sna>Brd^R* embryos. Analysis of fixed embryos actually indicated an increased elongation of the ventral cells in both *neur^degrad* and *sna>Brd^R* embryos (Fig. 3g; Supplementary Fig. 4o, p). How cell lengthening might be achieved in these embryos is studied below.

Finally, we addressed whether Neur indirectly regulates furrow invagination via its target Delta (Dl)[39,40]. Indeed, Neur regulates the endocytosis of Dl in the mesoderm[25,26,31] and this regulation is required for Notch receptor activation and mesectoderm specification[32]. As transcription of the *sna* gene is modulated by Notch in the mesoderm[41], we wondered whether the *neur^degrad* phenotype might result from a partial loss of Sna. Arguing against this possibility, we found that Sna was expressed normally in

*neur^degrad* embryos and that all Sna-expressing cells appeared to invaginate in *neur^degrad* embryos (Supplementary Fig. 4q, r). As loss of Dl activity in the female germline-blocked oogenesis[37], hence preventing us from looking at embryos lacking Dl, we studied *Notch*-mutant embryos derived from *Notch* germline clones. No furrow invagination defect was observed in these embryos (Supplementary Fig. 5a), indicating that Neur does not regulate gastrulation via Dl-Notch signaling. We next examined whether Neur acts via Stardust (Sdt), a recently identified target of Neur[42], and found that the loss of the Sdt isoforms that are targeted by Neur had no effect on furrow invagination (Supplementary Fig. 5b–g). Thus, Sdt is not the relevant target of Neur during gastrulation. Hence, we conclude that Neur regulates apical contractility by targeting proteins other than Dl and Sdt. These remains to be determined.

**Modeling predicts specific force changes in *neur* embryos**. We next wondered whether reduced apical contractility might be sufficient to account for the furrow defects seen upon Neur depletion and inhibition. We addressed this issue by resorting to computer simulation using the 2D multi-scale viscous model formulated by Conte et al.[43] We first performed a biomechanical analysis of furrow formation in wild-type embryo and next asked whether reducing apical forces is sufficient to reproduce in silico the defects associated with reduced Neur activity.

We assumed here that biomechanics of wild-type embryos is governed by cortex contractility of individual cells containing an incompressible viscous cytoplasm, enclosing a compressible viscous yolk and surrounded by a rigid vitelline membrane. In each cell, cortical contractility is contributed by forces along the apical, lateral, and basal edges (Supplementary Fig. 6a–e). The time-course distributions of these apical, lateral, and basal forces were previously obtained by means of video force microscopy (VFM)[44]. Although measured VFM forces were previously discretized in space and time[43], we used here the complete VFM data set[44] to build a spatial and temporal continuum of forces (Fig. 4a, d; Supplementary Fig. 6f, h–j). Of note, the resulting apical force profiles mirror the distribution of apical MyoII[15]. To increase computational stability, VFM force profiles were first symmetrized and regularized (Supplementary Fig. 6i–l). Also, since low-quality imaging of MyoII deep into the embryo likely under-estimated basal forces, we also tuned the topology of the VFM force distributions to best replicate in silico the gastrulation of wild-type embryos (see "Methods"). As a result, our model reproduced apical constriction, ventral cell lengthening and furrow invagination as measured in wild-type embryos (Fig. 4g, j, and Supplementary Movie 4; compare Fig. 4g with Supplementary Fig. 6g).

We then used computer simulations to explore what biomechanical transformations of the wild-type simulation might

**Fig. 4** In silico analysis of furrow invagination. **a–c** Spatial profiles of force distributions in wild-type (**a**), *neur* (**b**), and *Brd* (**c**) embryos represented as a one-dimensional epithelium at the bottom of each panel (color code: ectoderm, gray; mesectoderm, dark orange; lateral mesoderm, light orange; central mesoderm, yellow; dorsal is at ±180°, ventral at 0°). Apical forces (Gaussian red curve) and lateral forces (Sigmoid blue curve) are maximal in ventral cells whereas basal forces (green) are maximal in the ectoderm (at the indicated time points). Two time points are given for *Brd* embryos (**c**) to show the increase of apical forces in the ectoderm. The magnitude of the forces operating in silico is set by the viscosity of the cellular cytoplasm. For the sake of simplicity, the range of these forces was normalized per unit of viscosity. Consequently, forces range between 0 and 1, the latter being reached in the apical mesoderm. See Supplementary Fig. 6h–l for normalized experimental values in wild-type embryos. **d–f** Time evolution of force distribution in wild-type (**d**), *neur* (**e**), and *Brd* (**f**) embryos. To simulate *neur* embryos (**e**), apical (red) and lateral (blue) force distributions in the wild-type mesoderm (0°) were modified such that apical forces took longer to reach a maximum value weaker than their wild-type counterparts (shown in black) and that lateral forces were shifted in time relative to the wild-type ones (gray). To simulate *Brd* embryos (**f**), apical forces in the ectoderm (±180°) were increased to compete with those set in the mesoderm (0°) at furrow invagination (*t* = 21 min). **g–i** Time trends of apical constriction (central mesoderm area, blue; arbitrary units), furrow invagination (orange; arbitrary units) and ventral cell lengthening (green) in wild-type (**g**), *neur* (**h**), and *Brd* (**i**) embryos. See Supplementary Fig. 6g for experimental values. **j** Selected in silico views of wild-type, *neur*, and *Brd* embryos at various time points. Simulations reproduced the in vivo phenotypes. See Supplementary Movies 4 and 6

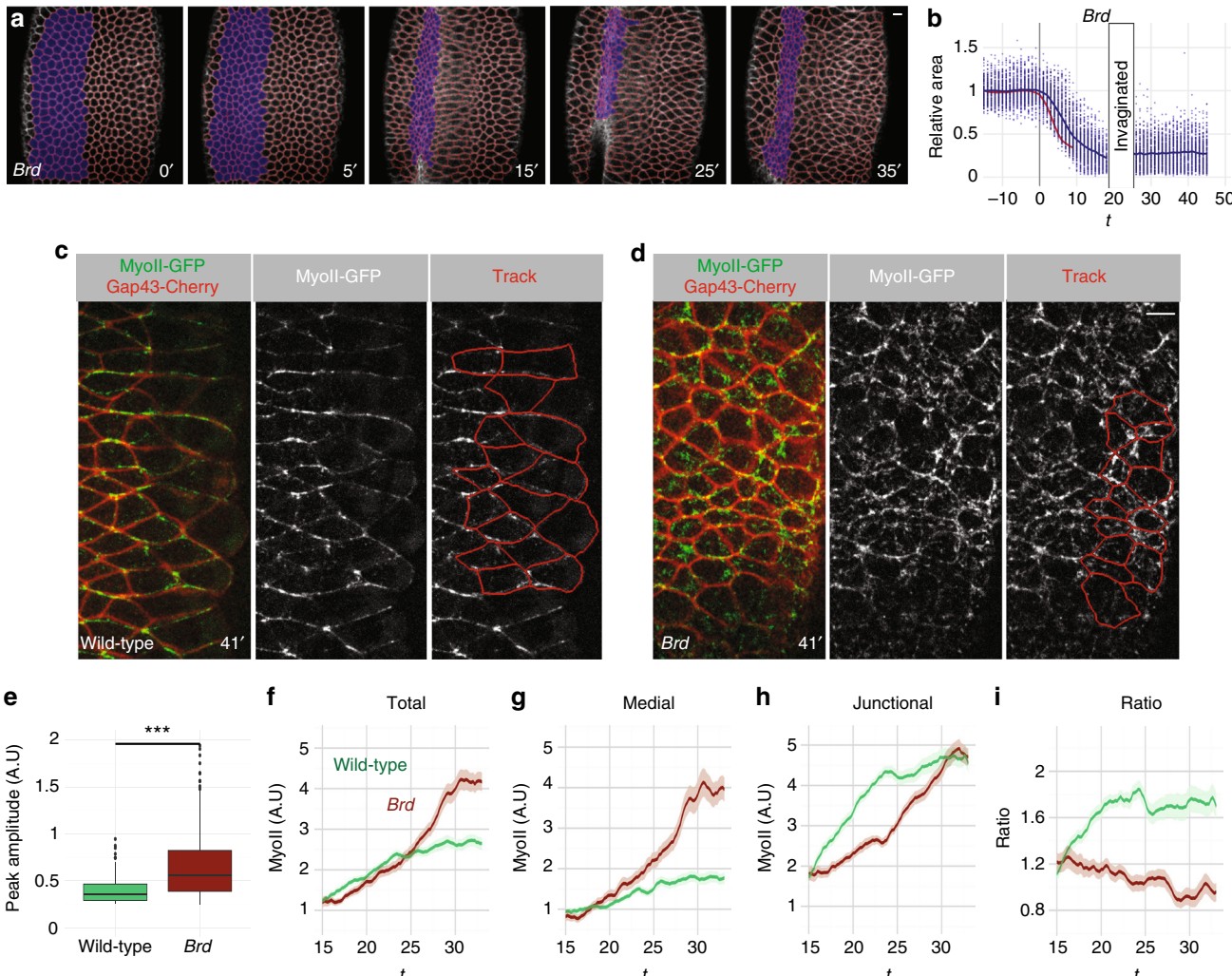

**Fig. 5** Furrow unfolding in *Brd* mutants. **a, b** Live imaging of a *Brd*-mutant embryo **a** expressing MyoII-GFP (not shown) and Gap43-Cherry (white; tracking in red). The furrow initially formed then regressed. Apical constriction (**b**, surface area values plotted as in Fig. 1f) was slightly slower in *Brd* mutants relative to wild-type (red curve). Once invaginated, ventral cells could no longer be tracked (**b**); they were tracked again after furrow unfolding. **c, d** Live imaging of MyoII-GFP (green) and Gap43-Cherry (red) in the ectoderm of wild-type (**c**) and *Brd* mutant (**d**) embryos. MyoII localized at junctions to form supracellular cables along the DV axis in wild-type embryos (ventral furrow, right). A medial–apical pool of MyoII could also be detected. Increased MyoII levels were detected at the medial–apical cortex in the ectodermal cells of *Brd*-mutant embryos (**d**). **e** Box plots of the amplitude values of the MyoII pulses measured at the medial–apical cortex of wild-type (n = 144 pulses measured in 50 cells from three embryos over a 24 min period) and *Brd*-mutant embryos (n = 350; 50 cells from three embryos over a 26.6 min period). MyoII levels were measured in lateral ectoderm cells (see tracking in **c, d**). Higher amplitude values levels were measured in *Brd* mutants (p < 10⁻¹⁵), showing that loss of *Brd* dramatically increased the level of MyoII contributing to assembly–disassembly pulses in the ectoderm. **f–i** Higher total and medial MyoII levels were measured in the ectoderm of *Brd*-mutant embryos (**f, g**; n = 3) relative to control embryos (n = 3). MyoII was mostly junctional in wild-type embryos (**h, i**)

induce a *neur^degrad* (or *sna>BrdR*) phenotype. Specifically, we investigated whether a slower increase of medosermal apical forces towards maximal values lower than those set for wild-type embryos, as suggested by our analysis of MyoII distribution (Fig. 2a, c, j), was sufficient to reproduce in silico the *neur* depletion/inhibition phenotype. We found that these changes were sufficient to reduce apical constriction and delay furrow invagination (Supplementary Fig. 7a–c). In this condition, however, the lengthening of the ventral cells appeared to be less pronounced than observed in *neur^degrad* and *sna>BrdR* embryos (Supplementary Fig. 7b; Fig. 3g). Therefore, guided by the proposed role of lateral forces in cell shortening[24], we investigated whether shifting in time lateral forces in the mesoderm might be sufficient to promote cell elongation. Simulations showed that decreasing apical forces while shifting in time lateral forces in the mesoderm produced embryos with reduced apical constriction,

delayed furrow invagination and increased cell lengthening (Fig. 4b, e, h, j). We therefore suggest that low Neur activity primarily results in reduced apical forces and, directly or not, delayed lateral forces in ventral cells.

**Furrow unfolding in *Brd*-mutant embryos.** As Neur positively regulates contractility, we next wondered whether the inhibition of Neur in the ectoderm by the Brd inhibitors might also contribute to furrow invagination. Consistent with a role of Brd in negatively regulating contractility in the ectoderm, we previously showed that apical MyoII was increased in the ectoderm of *Brd*-mutant embryos at stages 6–8[33]. However, no invagination defect was reported prior to disruption of epithelial polarity[33]. Here, we re-examined furrow invagination in *Brd*-mutant embryos using live imaging. We found that the ventral furrow invaginated but failed to completely close and eventually regressed as invaginated

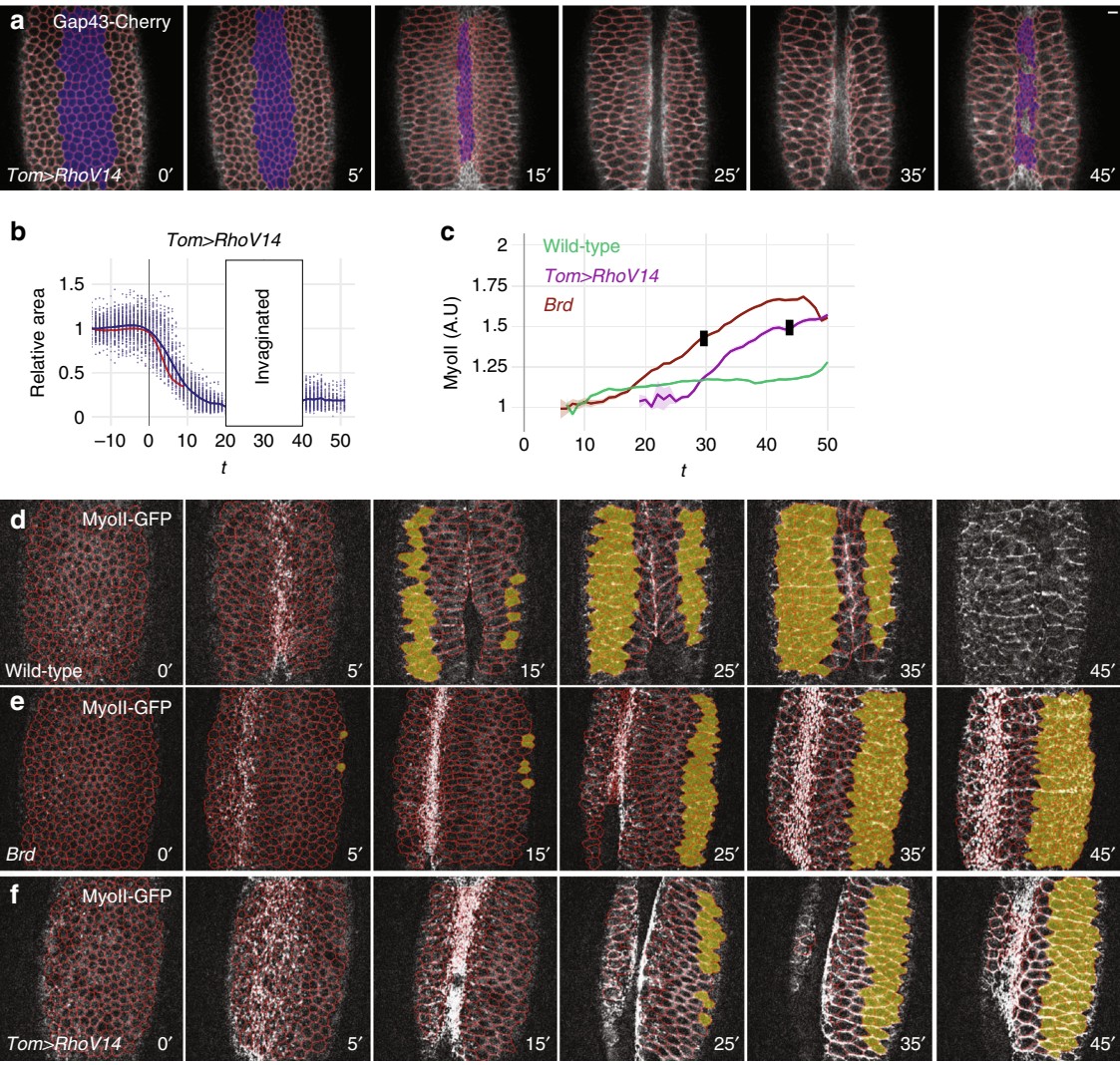

**Fig. 6** Increased contractility in the ectoderm is sufficient for furrow unfolding. **a**, **b** Snapshots from a *Tom>RhoV14* embryo movie (ventral view; Gap43-Cherry, white, and tracking, red). This embryo also expressed MyoII-GFP. While ventral furrow formation (**a**) and collective apical constriction (**b**; wild-type control, red) occurred normally, furrow unfolding was observed soon afterwards. **c** Mean MyoII intensity values measured in tracked ventral cells of wild-type (green), *Brd* (brown), and *Tom>RhoV14* (purple) embryos were plotted over time. Black stamps indicate the onset of furrow unfolding (mean values; see Supplementary Fig. 9e for the complete data set). **d–f** Snapshots from movies wild-type (**d**), *Brd* mutant (**e**), and *Tom>RhoV14* (**f**) embryos expressing MyoII-GFP (white) and Gap43-Cherry (tracking, red). Ventral–lateral views are shown. MyoII-GFP intensity was measured in ventral ectodermal cells (yellow-shaded in **d–f**) of wild-type (n = 3), *Brd* mutant (n = 3), and *Tom>RhoV14* (n = 3) embryos

cells appeared to be pulled out prior to mitosis (Fig. 5a, b; Supplementary Movie 5). This novel furrow unfolding phenotype indicates that the activity of *Brd* is required, presumably in the ectoderm, for complete furrow invagination. We further observed that the ventral furrow formed less rapidly in *Brd*-mutant embryos (Supplementary Fig. 8a) and that constricting ventral cells exhibited reduced anisotropy (Supplementary Fig. 8b). These observations raise the possibility that changes in the mechanical properties of the lateral cells in Brd mutants impact on how ventral cells constrict[20], reduce the speed of invagination and eventually trigger unfolding of the invaginating furrow.

We next tested in silico whether increasing apical forces in the ectoderm during furrow formation may be sufficient to unfold the ventral furrow. Using the 2D viscous model presented above, we found that a gradual increase of ectodermal apical forces was sufficient to block the complete invagination of the furrow (Fig. 4c, f, i, j; Supplementary Movie 6). Thus, increasing apical forces may cause furrow unfolding.

To examine whether apical forces are increased in the lateral cells of *Brd* mutants, we studied MyoII levels in the ectoderm. In wild-type embryos, MyoII accumulated apically in the ectoderm ~20 min after the onset of furrow invagination (Fig. 5c, f)[45]. Single cell tracking and quantification of MyoII intensity levels showed that MyoII predominantly accumulated at junctions where it formed multicellular cables oriented along the DV axis (Fig. 5c, h). MyoII was also detected at the medial–apical cortex, albeit at lower levels (Fig. 5g, i). Analysis of the temporal accumulation of MyoII indicated that medial–apical MyoII underwent few recruitment pulses of relatively low amplitude (Fig. 5e; Supplementary Fig. 8c, e). Analysis of *Brd*-mutant embryos showed that loss of Brd resulted in high levels of MyoII in the ectoderm (Fig. 5d, f). MyoII was primarily recruited at the medial–apical cortex, not at junctions, and this accumulation was associated with an increased number of MyoII pulses of higher amplitude (Fig. 5d–i; Supplementary Fig. 8d, f; Supplementary Movie 7). These changes in MyoII activity correlated with a reduction in size of the surface area in the lateral ectoderm

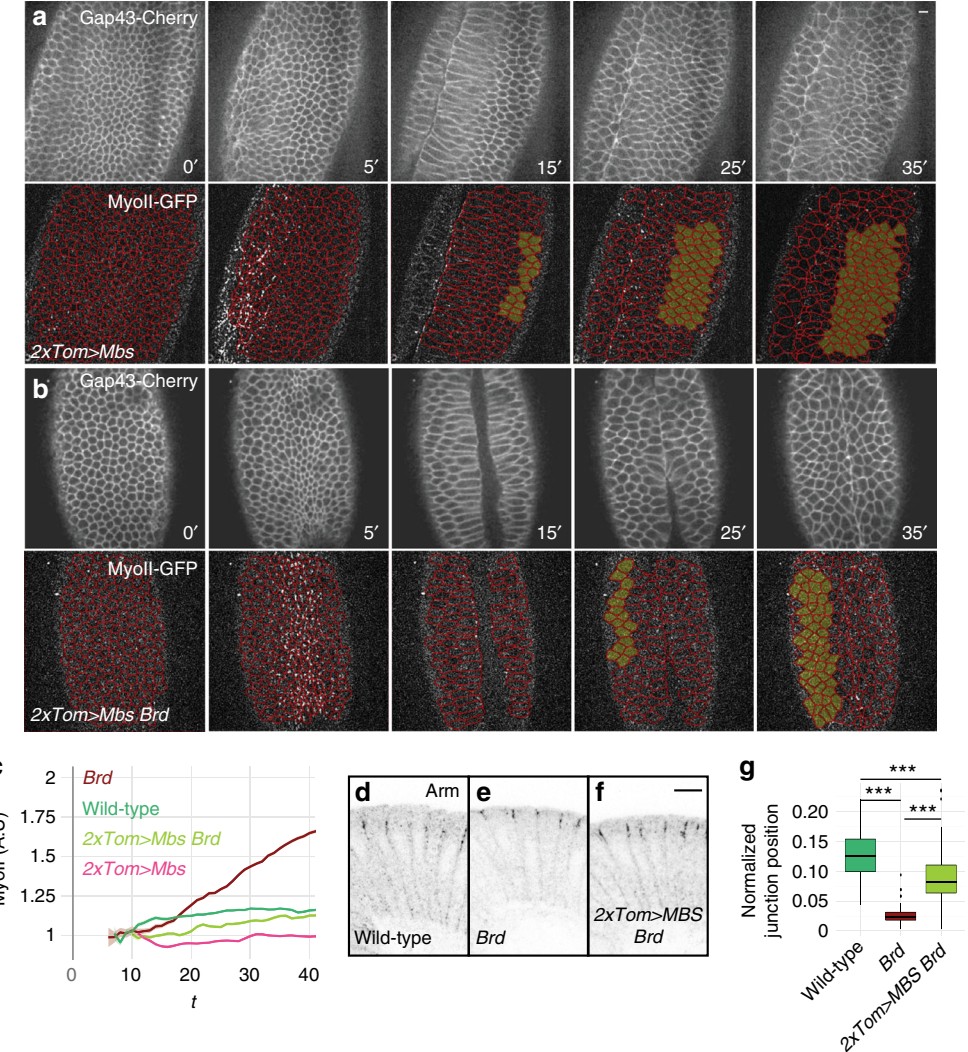

**Fig. 7** Decreased contractility in the ectoderm of Brd mutants suppresses furrow unfolding. **a, b** Live imaging of *2x Tom>Mbs* (**a**) and *Brd 2xTom>Mbs* (**b**) embryos expressing MyoII-GFP (bottom panels; tracking red) and Gap43-Cherry (top panels). Expression of MbsN300 in the ectoderm reduced MyoII levels in the ectoderm in both wild-type (**a**) and *Brd*-mutant embryos (**b**) and suppressed the *Brd*-mutant furrow unfolding phenotype. **c** Time-course accumulation of apical MyoII levels in ventral ectodermal cells of wild-type ($n = 3$), *Brd* ($n = 3$), *2xTom>Mbs* ($n = 3$), and *Brd 2xTom>Mbs* ($n = 3$) embryos. Ectopic Mbs in the ectoderm reduced MyoII levels in wild-type and *Brd*-mutant embryos. Mean and s.e.m. values are shown (see also Supplementary Fig. 10c, d and f). **d–g** The position of AJs (Arm, black) along the apical–basal axis was measured in the lateral cells of wild-type (**d**; 100 junctions, 7 embryos), *Brd* (**e**; 65 junctions, 5 embryos), and *Brd 2xTom>Mbs* embryos (**f**; 85 junctions, 6 embryos) at late stage 6. As shown earlier[33], AJ relocalize to the apical margin upon loss of *Brd* (as plotted in **g**; $p = 5 \times 10^{-10}$). This phenotype was suppressed by Mbs expression in the ectoderm (**g**; $p = 10^{-16}$)

(Supplementary Fig. 8g–i). These data strongly suggest that loss of Brd led to increased contractility in the ectoderm.

Taken together, simulations and experiments indicate that an increase in actomyosin levels at the apical cortex of ectodermal cells in *Brd*-mutant embryos likely renders the ectoderm less compliant to extension, which may in turn cause furrow unfolding. Thus, one activity of *Brd* is to limit the contractility of the apical cortex in the ectoderm.

**High rho activity in the ectoderm induced furrow unfolding.** To test whether increased contractility in the ectoderm may be sufficient to block furrow invagination, we induced the expression in the ectoderm of a constitutively active form of Rho1, RhoV14. RhoV14 was previously shown to increase actomyosin contractility in the early embryo[46,47]. The expression of RhoV14 was restricted to the ectoderm using the regulatory sequence of *Tom*, a *Brd* family gene repressed by Snail. Again, conditional expression was achieved by introducing a GFP reporter and a

transcriptional stop signal, flanked by two FRT sites, upstream of the *RhoV14* gene so that expression of RhoV14 in the mesoderm occurred only upon FLP-mediated excision of the stop cassette in the male germline (Supplementary Fig. 9a, b). Live imaging of *Tom>RhoV14* embryos indicated that increased Rho signaling in the ectoderm was sufficient to trigger furrow retraction following its invagination (Fig. 6a, b; Supplementary Movies 8, 9). Like in *Brd* mutants, we also observed that ventral furrow invagination was slowed down and that constricting ventral cells showed reduced anisotropy (Supplementary Fig. 9c, d), consistent with lateral cells resisting to the pulling force produced by ventral cells. These phenotypes correlated with increased apical MyoII levels in the ectoderm (Fig. 6c–f). Of note, MyoII accumulated mostly at junctions in *Tom>RhoV14* embryos (Supplementary Movie 8) and RhoV14 appeared to have a weaker effect than the *Brd* loss of function: it took longer for MyoII to reach similarly high levels and for the furrow to unfold (Fig. 6f; Supplementary Fig. 9e; see Supplementary Fig. 9f for an analysis of MyoII levels in the

mesoderm). Thus, increased MyoII activity in the ectoderm was sufficient to trigger furrow retraction in *Tom>RhoV14* embryos. As furrow retraction was observed at similar apical MyoII levels but different time points in *Tom>RhoV14* and *Brd*-mutant embryos (Fig. 6c), we suggest that ectoderm apical contractility reaches a threshold at which the ectoderm becomes less compliant to extension, which results in furrow unfolding. Thus, consistent with our interpretation of the *Brd*-mutant phenotype, furrow unfolding can result from increased contractility in the ectoderm.

**Low-ectoderm contractility suppressed furrow unfolding**. To further test this interpretation, we asked whether decreasing contractility in the ectoderm could be sufficient to suppress furrow retraction in *Brd*-mutant embryos. To do so, an active form of the myosin phosphatase Mbs, MbsN300[48], was specifically expressed in the ectoderm of *Brd*-mutant embryos using the same strategy as for RhoV14 (Supplementary Fig. 10a). To achieve different levels of Mbs expression, we studied embryos carrying one or two copies of the *Tom>Mbs* transgene (Supplementary Fig. 10b). Expression of Mbs in *1xTom>Mbs* and *2xTom>Mbs* embryos reduced apical MyoII levels in the lateral ectoderm from furrow invagination onwards (Supplementary Fig. 10c, d and Supplementary Movie 11; note that *2xTom>Mbs* had an earlier effect than *1xTom>Mbs* on MyoII levels in the ectoderm) but had no effect on MyoII accumulation and apical constriction in ventral cells during furrow invagination (Supplementary Fig. 10d).

We therefore used this experimental strategy to test the effect of decreasing contractility in the ectoderm of *Brd*-mutant embryos. Live imaging of *Brd*-mutant embryos expressing two copies of *Tom>Mbs* revealed that Mbs expression in the ectoderm was sufficient to reduce MyoII accumulation in the ectoderm and suppress furrow unfolding (Fig. 7a–c; Supplementary Movie 10). This showed that lowering MyoII activity in the ectoderm by promoting MyoII dephosphorylation is sufficient to counteract the increase in apical cortical contractility resulting from the loss of *Brd* activity. Of note, suppression was observed with *2xTom>Mbs* (n = 3/3) but not with *1xTom>Mbs* (n = 0/2). We therefore suggest that high levels of Mbs expression must be reached rapidly after furrow invagination in order to suppress the *Brd*-mutant phenotype.

Finally, we addressed whether the expression of Mbs could also suppress other aspects of the *Brd*-mutant phenotype, i.e., loss of epithelial integrity[33] and apical relocalization of Adherens Junctions (AJs) in the ectoderm[10,33,49]. We recently showed that the *Brd* epithelium polarity phenotype largely results from the downregulation of Sdt by Neur[42]. To test whether increased MyoII activity in the ectoderm of *Brd*-mutant embryos might also contribute to the loss of epithelium integrity, we studied the distribution of Patj, an apical polarity protein that directly interacts with Sdt. As reported previously[33], Patj failed to localize apically in *Brd*-mutant embryos at stage 7–8 (Supplementary Fig. 11a, b). We found here that expression of Mbs in *Brd* mutants weakly suppressed the loss of apical Patj localization in early embryos but clearly rescued epithelial polarity in the lateral non-neurogenic ectoderm (Supplementary Fig. 11c, d). This suggested that increased MyoII activity in *Brd* mutants contributes to the loss of epithelium integrity. We next tested whether Brd regulates AJ relocalization via its effect on Myosin, we asked whether this *Brd* phenotype could also be suppressed by the expression of Mbs in the ectoderm. To do so, we examined the position of AJs in stage 6 embryos. In *Brd*-mutant embryos, we confirmed that Armadillo (Arm; the fly β-catenin) localized at the apical margin in the lateral ectoderm, i.e., at a more apical

position compared to wild-type embryos[4,10,33]. Expression of Mbs in the ectoderm suppressed this localization defect (Fig. 7d–g). Thus, Brd likely modulates the myosin-dependent relocalization of AJs via its effect on apical MyoII activity. We conclude that the activity of *Brd* is required in the ectoderm to antagonize apical contractility and thereby delay AJ relocalization and propose that Neur may regulate AJ positioning in an indirect manner.

## Discussion

Our study uncovers a previously undescribed function for Neur that links DV patterning with contractility in the early embryo. Previous studies had indicated that the activity of Neur is restricted to the mesoderm via the transcriptional repression of the *Brd* family genes by Snail and the inhibition of Neur activity by Brd in the ectoderm[25,26,31]. Here, we provide several lines of evidence indicating that Neur acts in the mesoderm to regulate contractility. First, depletion of both maternal and zygotic Neur using the degradFP method perturbed collective apical constriction and slowed-down furrow invagination. Second, inhibition of Neur in the mesoderm by a stabilized version of Brd had similar effects, indicating that Neur acts in the mesoderm to promote ventral furrow formation. Moreover, ectopic Brd delayed the apical recruitment of MyoII and reduced the amplitude of the pulses of MyoII assembly–disassembly at the medial–apical cortex. Third, simulations indicated that lowering apical forces reproduced to a large extent the invagination defects seen upon depletion/inhibition of Neur. Thus, Neur is required in the mesoderm to regulate ventral furrow formation.

Interestingly, while our analysis of MyoII distribution at the apical–medial cortex strongly suggested that apical forces are both delayed and reduced in embryos with reduced Neur activity, simulations based on our 2D viscous model indicated that the phenotypes seen upon Neur depletion/inhibition, including ventral cell lengthening, may involve changes in the profile of cortical tensions that are required not only along the apical cortex but also at the lateral cortex. Specifically, in silico analysis indicated that the contractility of the lateral cortex is delayed in the mesoderm upon reduced Neur activity. Consequently, this raised the possibility that Neur acts in the mesoderm to also promote contractility along the lateral membranes. Of course, although models can prove that a certain physical mechanism is sufficient to explain an observed phenotype, they cannot prove that a certain mechanism is the reason that a phenotype is observed[50]. Since shortening of the ventral cells may result from a recoil of the stretched lateral cortex[24], a delayed increase of lateral forces may indirectly result from the slow elongation of the ventral cells. Nevertheless, the notion that Neur regulates the activity of MyoII at the lateral cortex is consistent with our analysis of AJ relocalization. Indeed, we found that the apical accumulation of AJs in the ectoderm of *Brd*-mutant embryos[33] was suppressed by the expression of Mbs. Since AJs move apically along the lateral cortex in a MyoII-dependent manner[49], this suppression supports a model whereby low Brd, i.e., high Neur, promotes the MyoII-dependent remodeling of AJs by regulating the activity of MyoII along the lateral cortex. Thus, our biomechanical model provides a useful framework to study how reduced Neur activity leads to defective furrow formation.

How Neur regulates the activity of MyoII in the mesoderm remains to be determined. As the two known targets of Neur, Dl and Sdt, are not involved in furrow formation, we propose that Neur acts via another molecular target. For instance, Neur might regulate the endosomal trafficking of the GPCRs Mist and/or Smog[8,13,51], or modulate the activity and/or levels of actomyosin

regulators. Future studies will aim at identifying the molecular targets of Neur in the mesoderm.

Our study further revealed that the Brd proteins are required in the ectoderm to allow furrow invagination. Live imaging indicated that *Brd*-mutant embryos had increased levels of MyoII in the ectoderm and exhibited increased contractility and decreased compliance to extension. Simulations indicated that corresponding force changes are sufficient to cause furrow regression. Furthermore, increasing Rho activity in the ectoderm of wild-type embryos was sufficient to trigger furrow unfolding and, conversely, decreasing MyoII activity in the ectoderm of *Brd*-mutant embryos was sufficient to suppress this phenotype. These findings strongly indicated that the levels of MyoII activity in the ectoderm need to be regulated and that the Brd proteins are critically required in the ectoderm to lower its contractility and allow proper furrow invagination. Although earlier computational studies had shown that the relative values of cortical tension between ventral and lateral cells were critical for proper furrow formation[7], this study provides the first experimental evidence showing that regulated contractility in the ectoderm is essential for furrow invagination. This finding further highlights that furrow invagination is an embryo-scale process[7,20,35]. Indeed, recent studies have revealed that the geometry of the constricting domain directs the shape of the invagination and regulates the anisotropy of constriction in each constricting cell[16,20,35,52]: constriction of a rectangular domain generates a furrow oriented along the long axis with each cell constricting more along the short axis whereas constriction of a square domain results in isotropic constriction. Thus, the organization of the actomyosin meshwork driving apical constriction depends on external mechanical constraints, used here as a readout of the geometry of the embryo. Consistent with this view, increased contractility in the ectoderm of *Brd*-mutant embryos, or upon expression of activated Rho in the ectoderm, changed the effective topology of the ventral contractile domain, mimicking a lateral broadening of this domain, and led to reduced anisotropy in constricting ventral cells.

In conclusion, Brd and Neur establish a biomechanical difference between the mesoderm and the ectoderm in the contractile activity of actomyosin. Thus, the Sna-Brd-Neur regulatory cascade regulates the topology of mechanical constraints at the embryo scale and links DV patterning with the spatial regulation of contractility.

## Methods

**Transgenes**. The pUASt-Brd-3xHA and pUASt-BrdR-3xHA transgenes were obtained by subcloning the Brd-3xHA and BrdR-3xHA DNA fragments—produced by PCR amplification and gene synthesis, respectively—into a pUASt-attB plasmid. All ten lysine (K) residues of the Brd open reading frame (ORF) were replaced by Arginine (R) in the BrdR sequence (primers and sequence can be obtained upon request). Following sequencing, both plasmids were integrated at the same PB[y+attP-9A] VK0013 (76A2) landing site.

The *sna>Brd^R* BAC transgene was generated using recombineering mediated gap-repair[53] starting from the attB-P[acman]-based BAC CH322-18i14[54]. The *snail* ORF (and 30 nt of the 5′ UTR) was replaced by a FRT-GFPnls-stop-FRT-BrdR-3xHA DNA fragment that was generated in a two-step PCR (primers and sequence can be obtained upon request). The resulting BAC was verified by sequencing of the recombined regions prior to phiC31-mediated integration at the PB[y+attP] VK00033 (65B2) landing site.

The *Brd-C* BAC[55] was modified by recombineering mediated gap-repair to produce the *Tom>RhoV14* and *Tom>Mbs* BACs. First, a 2.5 kb 5′ deletion (from −25,808 to −23,273 upstream the *Tom* ATG) and a 6.3 kb 3′ deletion (from +2734 to +9004 downstream the stop codon of the *Tom* gene) were introduced to remove the Bob and *Brd-Ocho* genes, respectively, leaving intact the gene and intergenic sequences of the *Tom* gene. Second, the ORF of *Tom* was replaced by a FRT-GFPnls-stop-FRT-RhoV14 and a FRT-GFPnls-stop-FRT-Mbs1-300 DNA fragment obtained by PCR amplification to produce the *Tom>RhoV14* and *Tom>Mbs* BACs, respectively. The sequences of the RhoV14 and Mbs1-300 genes were PCR amplified from genomic DNA prepared from UAS-RhoV14 (BL-8144) and UAS-Mbs1-300 (kind gift of J. Treisman) transgenic flies. The resulting BACs

were verified by sequencing of the recombined regions prior to phiC31-mediated integration at the PB[y+attP] VK00033 (65B2) and PB[y+attP] VK00020 (99F8) landing sites.

The GFP-Neur BAC was produced from CH322-162G17 BACs[54] was modified using recombineering mediated gap-repair to insert the EGFP flanked by GVG linkers after amino acid 46 in Neur-PA. This GFP-Neur BAC was integrated at the PB[y+attP-3B] VK00037 (22A3) site. A control Neur^WT BAC transgene was also produced by integrating the original CH322-162G17 BAC at the same landing site. Injection was performed by BestGene Inc (Chino Hills, CA).

**Flies**. To knock down the activity of *neur* in the female germline, we used the degradFP method[38]. Two copies of a mat-Gal4 transgene were combined with two copies of the UAS-NSlmb–VhhGFP4 in *neur* trans-heterozygous females carrying the *neur^IF65* and *neur^GE* mutations. We sequenced the strong hypomorphic *neur^IF65* allele and identified a A-to-T mutation in the splicing acceptor site at the cag/ttc (intron2/exon3) boundary. The lethality of these *neur^IF65*/*neur^GE* mutant females was rescued using a GFP-Neur BAC transgene (Supplementary Fig. 2a–c). Thus, females of the following genotype: PB[y + GFP-Neur]VK37 P[UAS-NSlmb–VhhGFP4]2/P[mat-Gal4]67; P[neo, FRT]82B *neur^GE* P[UAS-NSlmb–VhhGFP4]3/P[mat-Gal4]15P[neo, FRT]82B *neur^IF65* were crossed with male siblings and laid eggs maternally depleted of Neur. These embryos were referred to as *neur^degrad* embryos. As a control for this complex genotype, we used the same flies but rescued with the control Neur^WT BAC transgene. The progeny of these PB[y+Neur^WT]VK37 P[UAS-NSlmb–VhhGFP4]2/P[mat-Gal4]67; P[neo, FRT]82B *neur^GE* P[UAS-NSlmb–VhhGFP4]3/P[mat-Gal4]15P[neo, FRT]82B *neur^IF65* flies were referred to here as *control^degrad* embryos.

To study the complete loss of function of all eight *Brd* family members (BFM), we used the Df(3)Brd-C1 and Df(3)E(spl)δ-6 deletions[55]. These two deletions delete the *Brd*, *BobA*, *BobB*, *Tom*, *Ocho*, *E(spl)mα-BFM*, *E(spl)m4-BFM*, and *E(spl)m6-BFM* genes. Embryos homozygous for these two deletions are referred to here as *Brd*-mutant embryos. Note that the *E(spl)-HLH* genes of the *E(spl)-C* are also deleted in these embryos.

The UAS-Tom and UAS-mα transgenes used for over-expression are those used in ref. [33]. Embryos produced by P[UAS-mα]/+ or Y; P[mat-Gal4]67P[UAS-MyoII-GFP]/P[UAS-Tom]; P[mat-Gal4]15P[sqh-Gap43-Cherry]/+parents were studied by live imaging. The level of expression of Brd proteins should vary with genotype, i.e., transgene copy number, accounting for phenotypic variability. To circumvent this issue, we produced and studied the *sna>Brd^R* embryos.

The FRT-GFPnls-stop-FRT cassette was removed from the *sna>Brd^R*, *Tom>RhoV14*, and *Tom>Mbs* BAC transgenes using a FLP expressed in the male germline provided by the P{ry+betaTub85D-FLP}1 transgene (BL-7196). This resulted in a high frequency (80%) of FLP-mediated excision of the stop cassette in the male germline. These males were used in crosses with females expressing maternally the Gap43-mCherry and MyoII-GFP markers.

*Notch* GLC were produced in *N^55e11* FRT19A/ovo^D hs-flp FRT19A; P[sqh-Gap43-Cherry] /+females by heat-shocking (36.5 °C; 2 × 1 h) 24–48 and 48–72 h after egg laying (AEL) larvae. Embryos laid by females carrying visible *N*-mutant clones were studied by live imaging (the early *N*-mutant phenotype is not paternally rescued[32]). Embryos lacking the Neur-regulated isoforms of Sdt were *sdt^EH120* GLC-mutant embryos produced from *sdt^EH120* FRT19A/ovo^D hs-flp FRT19A; P[sqh-Gap43-Cherry]/+females or *sdt^Δ3-GFP*/Y-mutant embryos derived from *sdt^Δ3-GFP*/FM7 *ftz-lacZ*; P[sqh-Gap43-Cherry] P[sqh-MyoII-GFP] /+adult flies. Following imaging, *sdt^EH120*-mutant embryos were genotyped by single embryo PCR followed by sequencing (Supplementary Fig. 5c–e) and *sdt^Δ3-GFP*-mutant embryos were identified based on the absence of *lacZ*.

**Live imaging**. Embryos were dechorionated using bleach, glued on a slide, covered with halocarbon oil Voltalef 10S to avoid desiccation and flanked by a stack consistent of two 0.17 mm coverslips, on top of which a coverslip was added. Embryos were imaged at 22 °C using a Leica DMRXA microscope equipped with a CSU-X1 spinning disk, a back-illuminated Quantem 512C camera, 491 and 561 lasers and the Metamorph software.

Apical constriction was mostly studied on ($\Delta t = 1$ min, $\Delta z = 2$ μm, 11 z-sections) movies acquired using a 40× PlanApo NA = 1.40 objective. To study the actomyosin pulses, higher resolution movies were produced ($\Delta t = 12$ s, $\Delta z = 2$ μm, 6 z-sections) using a 63× PlanApo NA = 1.40 objective. In all movies, $t = 0$ corresponds to the onset of apical constriction, as determined visually on the Gap43-mCherry channel.

For segmentation, the Gap43-mCherry (membrane) signal was smoothed using the Fiji smoothing function and contrast was increased. To increase the number of cells that can be segmented in a single focal plane, a basal confocal section was chosen. Segmentation was then performed using Packing Analyser[56]. Quantification of the Myo-GFP signal was performed following maximum z-projection of all confocal sections, followed by a background subtraction using the available plugin in Fiji. Since the MyoII signal is maximal apically, this allowed us to analyze MyoII in cells that are slightly peripheral in the field of view. To do so, the segmentation mask produced using Packing Analyser was superposed on top of the MyoII-GFP images and the average signal intensity was measured from the raw GFP signal in each segmented cell using Fiji. For each movie, measured values were normalized relative to the initial values (corresponding to the GFP signal measured prior

to the apical recruitment of MyoII; this signal varied with montage conditions). To measure the medial–apical pool of MyoII, the segmentation mask was eroded 3 times using the erode FIJI function, thereby creating an exclusion mask of 3-pixel wide (pixel size is in µm). The junctional pool was determined by substracting the medial pool of MyoII from the total pool. Pulse analysis was performed on the medial pool of MyoII. For each cell, local minima were automatically identified, thereby defining local peaks that were fitted as Gaussian peaks above a linear background using:

$$(x) = a0 \times x + b0 + a1 e^{-\left(\frac{x-b1}{c1}\right)^2},$$

where $a1$ is the peak amplitude, $b1$ the peak centroid, and $c1$ is related to the peak width. The $a0$ and $b0$ values correspond to linear background values. The function *lsqcurvefit* function of R-*pracma* package was used to define peaks. Automatic fitting was manually corrected. Peaks with amplitude values inferior to 25% of the initial value were discarded.

All graphics and data were analyzed using R studio.

Wild-type controls were embryos maternally loaded with MyoII-GFP and Gap43-mCherry, i.e., laid by P[psqh-MyoII-GFP], [psqh-Gap43-mCherry]/+females. The presence of the *Brd* deficiencies at the homozygous state was unambiguously determined after imaging by single embryo PCR. Similarly, for all experiments involving the *sna>BrdR*, *Tom>RhoV14*, and *Tom>Mbs* BAC transgenes, the excision of the FRT-GFPnls-stop-FRT cassette was determined after imaging using single embryo PCR. Embryos were allowed to develop overnight at 18 °C prior to DNA extraction using Tris pH 8.2 10 mM, EDTA 1 mM, NaCl 25 mM, and Proteinase K 200 µg/ml. All primers used for embryo genotyping are listed in the Supplementary Table 1.

For each genotypes, several properly oriented embryos were studied: wild-type ($n = 7$ at 40×, Figs. 2a and 6c; $n = 3$ at 63×, Fig. 2g; $n = 2$ at 63×, Fig. 5e–i; $n = 7$, Fig. 7d, g), *mat>Tom+mα* ($n = 9$ at 40×, Fig. 2c; $n = 6$ at 63×, Fig. 2g), *sna>BrdR* ($n = 5$ at 40×, Fig. 2a), *neur^degrad* ($n = 19$ at 40×, Fig. 3f), *control^degrad* ($n = 6$ at 40×, Supplementary Fig. 4), *Brd* ($n = 5$ at 40×, Figs. 5b and 6c; $n = 3$ at 63×, Fig. 5d; $n = 5$, Fig. 7e, g), *Tom>RhoV14* ($n = 3$ at 40×, Fig. 7a–c), *Brd 2xTom>Mbs* ($n = 5$ at 40×, Fig. 7b, c; $n = 6$, Fig. 7f, g), *sdt^EH120* ($n = 2$ at 40×, Supplementary Fig. 5), and *sdt^Δ3GFP* ($n = 5$ at 40×, Supplementary Fig. 5).

**Immunostaining and western blots.** For cross sections, dechorionated embryos were heat-fixed (10 s in boiling water with 0.4% NaCl and 0.03% Triton X-100). These were then incubated with primary antibodies in PBS1x Triton 0.1% for 2 h at 25 °C (or overnight at 4 °C), washed and incubated with secondary antibodies in PBS1x Triton 0.1% for 2 h at 25 °C (or overnight at 4 °C). Embryos were then post-fixed in PBS1x with 4% PFA for 1 h. Embryos of the proper stage were then selected under the binocular scope, cut using a sharp scalpel blade and mounted in 4% N-propyl-galate, 80% glycerol. Ovaries were fixed 20 min with 4% PFA in PBS and stained for 2 h with Phalloidin-atto647N (1:1000; Sigma #65906) in PBS triton 0.1%. Wild-type controls were *w^1118* embryos and ovaries.

The following primary antibodies were used: Armadillo (mouse N2 7A1 mAb; 1:500; DSHB), cytoplasmic MyoII (rabbit; 1:500; gift from D. Kiehart), intracellular Dl (rat anti-DlICD mAb 10D5; 1:1000; gift from M. Rand), GFP (goat; 1:500; Abcam #6673), Neurotactin (Nrt; mouse anti-Nrt mAb BP106; 1:500; DSHB), Twist (rabbit; 1:500; gift from M. Eisen), Snail (rabbit; 1:500; gift from M. Eisen), V5 (rabbit; 1:500; Sigma #V8137), and Patj (guinea-pig; 1:500; gift from M. Krahn).

Images were acquired using a confocal Zeiss LSM780 microscope with a 63× (PL APO, N.A. 1.4 DIC M27) objective.

For each genotype, several embryos were imaged: wild-type (25 sections from 4 experiments, Figs. 2h and 3c), *sna>BrdR* (13 sections from 2 experiments, Fig. 2i), *neurGFP* (6 sections from 1 experiment, Fig. 3a), *neur^degrad* (25 sections from 2 experiments, Fig. 3d), *sna>GFP* (5 sections from 1 experiment, Supplementary Fig. 1g), and *Tom>GFP* (5 sections from 1 experiment, Supplementary Fig. 6e).

The position of AJs relative along the apical–basal axis was determined on sectioned embryos stained with anti-Arm by measuring the distance between the embryo surface and the Arm signal on five junctions per section. Analysis of cell lengthening was also performed on sections: the height of ventral cells was measured and normalized relative to the height of dorsal ectoderm cells. Staging was based on furrow invagination (stage I and II, as defined by Sweeton et al.[5])

V5-tagged Mbs was studied by western blot analysis. Protein extracts were prepared from embryos and loaded on 4–20% precast Miniprotean TGX gels GEL for SDS-PAGE. Proteins were transferred onto 0.2 µm Nitrocellulose membranes (Biorad). Rabbit anti-V5 (Sigma, V8137), mouse anti-tubulin (DM1A, Sigma), and HRP-coupled secondary antibodies (GE Healthcare) were used for detection with SuperSignal WestFemto (Thermo Scientific). The uncropped western blot is shown in Supplementary Fig. 10.

**Modeling.** A multi-scale 2D finite-element model was used to study the bio-mechanics of early embryos at the subcellular, cellular, and epithelial scales. As the physical–mathematical framework of the model is described in detail in Conte et al.[43] and Brodland et al.[57], we only focus here on its key features. The initial configuration of the in silico model of the embryo consists of a circular epithelium of 84 cells (Supplementary Fig. 6a–e). The finite-element model of each cell takes into account the subcellular effects of the complex cellular force fields by assuming

that these can be resolved into equivalent net forces along cell edges surrounding a viscous material representing the inner cytoplasm. The cytoplasm was assumed to be incompressible which was simulated by enforcing that the surface enclosed by each cell membrane remain constant over time. To grant higher degrees of cell deformation, membranes were subdivided into 5 sections along the apical–basal axis and three sections along the apical and basal sides (Supplementary Fig. 6a, b). Thus, each cell comprises 15 finite elements that emulate the viscous response of the cell locally[57] (Supplementary Fig. 6c). The epithelium was assumed here to respond viscously to deformations. This assumption is consistent with recent evidence indicating that during the lengthening phase of ventral furrow formation the entire bulk of the epithelial tissue below the apical cortex behaves viscously[23]. Although the epithelium is unlikely to be elastic on the time scales of the early ventral furrow formation, the extent to which the elastic response of the embryo matters at the time scales of the invagination process remains to be determined[52,58–60]. The viscosity of the cytoplasm was further assumed to be equal to the one of the yolk. Although it is still not clear whether the yolk volume remains constant throughout gastrulation[59], volume conservation in 3D does not imply conservation in a 2D section. For this reason, variations in the surface of the yolk were permitted (<6% of the initial area), an assumption which we will refer to as 2D yolk compressibility. Also, as the yolk squirts out when the embryo is pierced during gastrulation[59], we modeled the yolk as a viscous fluid exerting a pressure on the basal side of the epithelium to a maximum of 10% of its initial value. The value of this variable was chosen so that surface variations of the yolk were similar to those observed in vivo[43]. Finally, the vitelline membrane was modeled as a rigid boundary condition. As a result, forces act along the edges of individual cells to drive the viscous motion of nodes against an incompressible cytoplasm and a 2D compressible yolk within a fixed cross-section area (Supplementary Fig. 6a–e). As the magnitude of the forces operating in silico is set by the viscosity of the cyto-plasm in this context, forces were normalized per unit of viscosity. Consequently, forces range between 0 and 1, the latter being reached in the apical mesoderm in wild-type embryos (Fig. 4a, d).

We next used the complete in vivo force data set that was previously obtained through the VFM method[43] to define the spatial-temporal distributions of forces governing ventral furrow formation in the wild-type embryo (Supplementary Fig. 6f–h). To take into account the DV symmetry of our model and to avoid numerical instabilities, these VFM force distributions were symmetrized and regularized. Several modifications of the resulting force distributions were further introduced to better replicate in silico furrow invagination as seen in vivo: (i) the ventral domain of lateral cortex contraction was adjusted so that it could reach maximum value over the whole central mesoderm; (ii) the ectodermal basal forces were allowed to more gradually fade towards the ventral midline; (iii) the time trends of VFM forces were linearized, which allowed us to study other possible timing configurations (Fig. 4a, d; Supplementary Fig. 6h–l).

**Repeatability.** Result repeatability is supported by the number of embryos, cells, sections, and images that have been analyzed (see Figures, Results, and Method sections) and key limitations are discussed in the text (see Discussion of the *mat>Tom+ma* and *neur^degrad* genotypes). In our experiments, the size of the effect associated with our experimental perturbations were not pre-specified. Instead, we aimed for the highest spatial and temporal resolution, strongest signal-to-noise ratio, and tightest control in conditional gene perturbations that are currently permitted.

**Data availability.** The authors declare that all data supporting the findings of this study are available within the article and its Supplementary information files or from the corresponding author upon reasonable request.

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

## Acknowledgements

We thank M. Affolter, B. Aigouy, E. Caussinus, M. Eisen, D. Kiehard, E. Knust, M. Krahn, A. Martin, M. Rand, J. Treisman, the Bloomington *Drosophila* Stock Center, the Developmental Studies Hybridoma Bank (DSHB), and Flybase for flies, antibodies and

other resources. We thank S. Chanet for her initial input, G.W. Brodland for his input and support in the biomechanical modeling, D. del Alamo for molecular analysis of the neur[IF65] allele, L. Couturier for technical help, and M. Labouesse for critical reading. This work was funded by an ANR grant (ANR12-BSV2-0010-01 to F.S.) and by MINECO grants (BFU2016-75101-P and RYC-2014-15559 to V.C.). V.C. was also supported by the Centro de Excelencia Severo Ochoa Award to the Institute of Bioengineering of Cata-lonia. G.P.-M. was supported by MENRT and FRM fellowships.

## Author contributions

G.P.-M. performed the experiments and the quantitative image analysis. V.C. performed the modeling. K.M. generated the BAC transgenes. V.R. and G.C. provided technical help with embryo analysis. G.P.-M., V.C., and F.S. designed the study, analyzed the data, and wrote the paper.

## Additional information

**Competing interests:** The authors declare no competing financial interests.

