## [Peer Review File · Nature Communications]

Reviewers' Comments:

Reviewer #2:

Remarks to the Author:

This is a generally well-done review. I am still unsure about the localization of myosin in neur[degradFP] embryos. Looking at the new data in Supplementary Figure 4l, m, it still seems to me that the myosin II signal (red) spreads over a greater number of cells in neur[degradFP] embryos than in the wild type. I suggest that the authors do measure the number of cells that accumulate apical actin in wild-type and neur[degradFP] embryos to clearly demonstrate that in neur[degradFP] there is not an expansion of the pool of ventral cells that accumulate myosin apically. The Snail analysis in Supplementary Figure 4o, p alleviates some of my concerns with the specification of the mesoderm in neur[degradFP] embryos, but I still wonder about a possible change in width of the myosin II domain comparing Supplementary Figure 4l and m.

Reviewer #3:

Remarks to the Author:

The authors have provided substantial new data in response to reviewers' suggestions and have carefully addressed many of the concerns. I believe the findings would be of great interest to the field of tissue patterning and morphogenesis and therefore strongly recommend publication of this nice work.

I do however have a few suggestions and final comments:

1. The paragraphs on page 13 sound a bit random in a section titled "Lowering contractility in the ectoderm suppressed furrow unfolding." It would read better if they were included in a separate section focusing on the role of Brd in epithelial polarity and adherens junctions, with its own section title (such as "Lowering contractility partially suppressed the Brd epithelium integrity phenotype").

2. The authors did many experiments trying to address how Neur promotes apical myosin recruitment, but these experiments were hardly mentioned in their revised text. Although the experiments did not reveal the target of Neur, I feel the results would still be informative. The short paragraph (the 2nd paragraph) on page 15 could be further expanded to include the considerable attempts that the authors made to try to identify the target of Neur.

3. Fig 3g-j: One potential caveat of measuring cell length in fixed embryos is that it is unclear whether the stages of the embryos exactly match to each other. During the initial phase of ventral furrow formation, the length of the constricting cells increases by ~70% within 10-12 minutes, so one can imagine that a slight mismatch on stages may result in a substantial difference in cell length. There is also no direct evidence for the existence of a lateral shortening force in the mesoderm cells. The proposed model that the onset of the lateral force is delayed in neur depletion/inhibition embryos is interesting but seems to remain speculative.

4. The observation that the intensity of apical myosin in the Brd mutant and the Tom>RhoV14

mutant is higher than that of wildtype (Fig. 6c, 6d, and 6e) is intriguing. The new quantification the authors made and presented in the rebuttal letter in my opinion is worth to present in the paper.

A few minor points:

The resolution of the plots in Fig S1a and b are very low.

Fig 1f-h, Fig3f: the authors should indicate the number of embryos quantified for each plot.

Figure legend 2f: "(f). Moreover, higher amplitude values levels were measured": "levels" should be deleted.

Supplemental figure legend 8: last line: " $p=3 \cdot 10^{-6}$ " should be " $p=3 \times 10^{-6}$ ".

Reviewer #2 (Remarks to the Author):

This is a generally well-done review. I am still unsure about the localization of myosin in neur[degradFP] embryos. Looking at the new data in Supplementary Figure 4l, m, it still seems to me that the myosin II signal (red) spreads over a greater number of cells in neur[degradFP] embryos than in the wild type. I suggest that the authors do measure the number of cells that accumulate apical actin in wild-type and neur[degradFP] embryos to clearly demonstrate that in neur[degradFP] there is not an expansion of the pool of ventral cells that accumulate myosin apically. The Snail analysis in Supplementary Figure 4o, p alleviates some of my concerns with the specification of the mesoderm in neur[degradFP] embryos, but I still wonder about a possible change in width of the myosin II domain comparing Supplementary Figure 4l and m.

We appreciate this point. Indeed, the apical area with apical MyoII appeared to be larger in neur[degradFP] embryos than in wild-type embryos. This difference is clearly contributed by the difference in apical constriction (which can be misleading when correlating area with cell number). It might also be contributed, as suggested by the referee, by a difference in the number of cells accumulating MyoII apically. However, we could not precisely count the number of cells accumulating MyoII on sections. A precise counting would require 2-color movies of neur[degradFP] embryos (to track cells and measure MyoII) which is not technically possible with current markers (due to the anti-GFP nanobodies). Thus, we cannot provide precise cell numbers. While we cannot exclude that more cells accumulate apical MyoII in neur[degradFP] embryos, we note that no change in the spatial pattern of MyoII accumulation was seen in sna>BrdR embryos (see Figure 3j). For this reason, we do not think that reduced Neur has a major impact on the number of cells accumulating apical MyoII.

Reviewer #3 (Remarks to the Author):

The authors have provided substantial new data in response to reviewers' suggestions and have carefully addressed many of the concerns. I believe the findings would be of great interest to the field of tissue patterning and morphogenesis and therefore strongly recommend publication of this nice work. I do however have a few suggestions and final comments:

1. The paragraphs on page 13 sound a bit random in a section titled "Lowering contractility in the ectoderm suppressed furrow unfolding." It would read better if they were included in a separate section focusing on the role of Brd in epithelial polarity and adherens junctions, with its own section title (such as "Lowering contractility partially suppressed the Brd epithelium integrity phenotype"). we have re-organized the last two paragraphs. We hope they read better.

2. The authors did many experiments trying to address how Neur promotes apical myosin recruitment, but these experiments were hardly mentioned in their revised text. Although

the experiments did not reveal the target of Neur, I feel the results would still be informative. The short paragraph (the 2nd paragraph) on page 15 could be further expanded to include the considerable attempts that the authors made to try to identify the target of Neur.

Since these results are negative in nature, we feel that these would not be so useful.

3. Fig 3g-j: One potential caveat of measuring cell length in fixed embryos is that it is unclear whether the stages of the embryos exactly match to each other. During the initial phase of ventral furrow formation, the length of the constricting cells increases by ~70% within 10-12 minutes, so one can imagine that a slight mismatch on stages may result in a substantial difference in cell length. There is also no direct evidence for the existence of a lateral shortening force in the mesoderm cells. The proposed model that the onset of the lateral force is delayed in neur depletion/inhibition embryos is interesting but seems to remain speculative.

We understand this concern. To address this point, we now show all the embryos studied in these experiments, from stage I to stage III in the Supplementary Figure 4o,p. Our results show that this difference in cell lengthening is also seen in stage II embryos, prior to cell shortening at stage III. Thus, a staging mismatch cannot explain the difference in cell lengthening observed between wild-type and experimental embryos.

4. The observation that the intensity of apical myosin in the Brd mutant and the Tom>RhoV14 mutant is higher than that of wildtype (Fig. 6c, 6d, and 6e) is intriguing. The new quantification the authors made and presented in the rebuttal letter in my opinion is worth to present in the paper.

We now present these results in Supplementary Figure S9f.

A few minor points:

The resolution of the plots in Fig S1a and b are very low.

fixed

Fig 1f-h, Fig3f: the authors should indicate the number of embryos quantified for each plot.

done

Figure legend 2f: "(f). Moreover, higher amplitude values levels were measured": "levels" should be deleted.

deleted

Supplemental figure legend 8: last line: "p=3 10-6" should be "p=3x10-6".

changed